# Rare Deletions or Large Duplications Contribute to Genetic Variation in Patients with Severe Tinnitus and Meniere Disease

**DOI:** 10.3390/genes15010022

**Published:** 2023-12-22

**Authors:** Alba Escalera-Balsera, Alberto M. Parra-Perez, Alvaro Gallego-Martinez, Lidia Frejo, Juan Martin-Lagos, Victoria Rivero de Jesus, Paz Pérez-Vázquez, Patricia Perez-Carpena, Jose A. Lopez-Escamez

**Affiliations:** 1Otology & Neurotology Group CTS495, Instituto de Investigación Biosanitaria, ibs.GRANADA, Universidad de Granada, 18071 Granada, Spain; albaescalera@correo.ugr.es (A.E.-B.); amparraperez@ugr.es (A.M.P.-P.); alvaro.gallego@ugr.es (A.G.-M.); lidia.frejonavarro@sydney.edu.au (L.F.); juan.martinlagos.sspa@juntadeandalucia.es (J.M.-L.); percarpena@ugr.es (P.P.-C.); 2Division of Otolaryngology, Department of Surgery, Universidad de Granada, 18016 Granada, Spain; 3Sensorineural Pathology Programme, Centro de Investigación Biomédica en Red en Enfermedades Raras, CIBERER, 28029 Madrid, Spain; 4Meniere’s Disease Neuroscience Research Program, Faculty of Medicine & Health, School of Medical Sciences, The Kolling Institute, University of Sydney, Sydney, NSW 2065, Australia; 5Department of Otorhinolaryngology, Hospital Clinico Universitario San Cecilio, 18016 Granada, Spain; 6Department of Otorhinolaryngology, Hospital Clínic, 08036 Barcelona, Spain; vriveroj@csi.cat; 7Servicio de Otorrinolaringología, Hospital Universitario Central de Asturias, 33011 Oviedo, Spain; pazperezv@gmail.com; 8Department of Otorhinolaryngology, Hospital Universitario Virgen de las Nieves, 18014 Granada, Spain

**Keywords:** Meniere disease, tinnitus, genomics, structural variant, bioinformatics

## Abstract

Meniere disease (MD) is a debilitating disorder of the inner ear defined by sensorineural hearing loss (SNHL) associated with episodes of vertigo and tinnitus. Severe tinnitus, which occurs in around 1% of patients, is a multiallelic disorder associated with a burden of rare missense single nucleotide variants in synaptic genes. Rare structural variants (SVs) may also contribute to MD and severe tinnitus. In this study, we analyzed exome sequencing data from 310 MD Spanish patients and selected 75 patients with severe tinnitus based on a Tinnitus Handicap Inventory (THI) score > 68. Three rare deletions were identified in two unrelated individuals overlapping the *ERBB3* gene in the positions: NC_000012.12:g.56100028_56100172del, NC_000012.12:g.56100243_56101058del, and NC_000012.12:g.56101359_56101526del. Moreover, an ultra-rare large duplication was found covering the *AP4M1*, *COPS6*, *MCM7*, *TAF6*, *MIR106B*, *MIR25*, and *MIR93* genes in another two patients in the NC_000007.14:g.100089053_100112257dup region. All the coding genes exhibited expression in brain and inner ear tissues. These results confirm the contribution of large SVs to severe tinnitus in MD and pinpoint new candidate genes to get a better molecular understanding of the disease.

## 1. Introduction

Tinnitus is the perception of an internal sound in the absence of external stimulation and affects 15% of the population. It is usually associated with sensorineural hearing loss (SNHL) [1]. It is also a clinical symptom associated with Meniere disease (MD, OMIM 156,000 [2]), a chronic disorder of the inner ear defined by episodes of vertigo associated with fluctuating SNHL and tinnitus [3].

Twins [4], adoptees [5], and familial aggregation studies [6] support tinnitus heritability, particularly for severe and bilateral tinnitus [4]. Genetic variation in the DNA sequence can explain this heritability across different tinnitus phenotypes, supporting a multiallelic model of inheritance consisting of common and rare variants that exert additive effects on the trait [7]; however, interpreting non-coding and missense variants remains challenging.

Severe tinnitus is experienced by 1–2% of the population and has been defined as “Tinnitus Disorder”. It is associated with hyperacusis, anxiety with emotional distress, and cognitive dysfunction, leading to functional disability and behavioral changes [8].

In Spanish patients with MD, using exome sequencing and gene burden analyses, severe tinnitus has been associated with rare missense variants in 24 synaptic genes, including *ANK2*, *TSC2*, and *AKAP9*. This finding was replicated in a Swedish cohort of tinnitus patients without MD, supporting the significant role of rare single nucleotide variants (SNVs) with a strong effect size [9].

Conversely, the contribution of structural variants (SVs) has seldom been investigated in the genetic structure of tinnitus. Gallego-Martinez et al. [10] recently reported several ultra-rare SVs in highly constrained regions in Swedish patients with tinnitus.

Furthermore, the functional role of large heterozygous SVs and copy number variants (CNVs) involving one or several genes deserves more attention since they may explain: (a) the cumulative effect of intermediate-impact non-coding variants of unknown significance (VUS) on the phenotype and (b) the concurrence of comorbidities in complex phenotypes associated with a pleiotropic effect [11].

This study aimed to investigate SVs in MD patients and patients with severe tinnitus through exome sequencing datasets. We report four CNVs in patients with MD and severe tinnitus: three rare deletions in the *ERBB3* gene, and a large duplication involving *AP4M1*, *COPS6*, *MCM7*, and *TAF6*.

## 2. Materials and Methods

### 2.1. Patient Selection

In this work, 310 definite MD patients were enrolled from Spanish referral centers through the MD Consortium. They were diagnosed with definite MD according to the diagnostic criteria described by the International Classification Committee for Vestibular Disorders of the Barany Society [12]. Tinnitus Handicap Inventory (THI) questionnaires were used to assess the impact of tinnitus on health-related quality of life, and pure-tone audiograms were used to evaluate SNHL. The THI score distribution and air-conducted hearing thresholds were represented using ggplot2 [13], tidyr [14], ggpubr [15], dplyr [16], and scales [17] R packages.

The THI score did not follow a normal distribution; thus, patients were classified into four subgroups according to their quartiles. Individuals with a THI score higher than the third quartile score were categorized as having severe tinnitus, whereas those below this threshold were categorized as having non-severe tinnitus.

This study was carried out according to the principles of the Declaration of Helsinki, revised in 2013 [18]. The human ethics research protocol was approved by the Institutional Review Board (Protocol number: 1805-N-20), and all the subjects signed a written informed consent form to donate biological samples for genetic studies.

### 2.2. DNA Sequencing and Dataset Generation

Blood samples were obtained from each patient to isolate the DNA and sequence the exome. DNA extraction from whole blood was carried out using the QIAamp DNA Blood Mini Kit (Qiagen, Hilden, Germany), following the manufacturer’s protocol. The concentration and quality of DNA was assessed using Nanodrop (Thermo Fisher, Waltham, MA, USA) and Qubit (Invitrogen, Waltham, MA, USA), as previously outlined [19]. Additionally, DNA integrity was confirmed through electrophoresis in a 2% agarose gel. For exome sequencing, the minimum accepted parameters included a concentration exceeding 20 ng/µL, a 260/280 ratio exceeding 1.8, and the absence of observable smearing or DNA degradation during electrophoresis.

The standard exome capture libraries were generated utilizing the SureSelectXT Human All Exon V6 kit (Agilent Technologies, Santa Clara, CA, USA), employing 1 µg of input genomic DNA (gDNA) for each sample. The gDNA was diluted with EB Buffer and sheared to achieve a target peak size of 150–200 bp using the Covaris LE220 focused-ultrasonicator (Covaris, Woburn, MA, USA), following the manufacturer’s recommendations. Subsequently, end-repair and the addition of an “A” tail were executed, followed by the ligation of Agilent adapters to the fragments. Following the assessment of ligation efficiency, the adapter-ligated product underwent PCR amplification. After the final product was purified, quantification was performed using TapeStation DNA screentape D1000 (Agilent Technologies, Santa Clara, CA, USA Agilent). Then, 250 ng of the DNA library was combined with hybridization buffers, blocking mixes, RNase block, and 5 µL of the SureSelect all exon capture library, following the standard Agilent SureSelect Target Enrichment protocol. The captured DNA underwent washing and amplification. Finally, the purified product was quantified via qPCR in accordance with the qPCR Quantification Protocol Guide (KAPA Library Quantification kits for Illumina Sequencing platforms) and was assessed using TapeStation DNA screentape D1000.

The binary base calling files produced by the Illumina platform, using the integrated primary analysis known as RTA (real-time analysis), were transformed into the FASTQ file format using the Illumina package bcl2fastq v2.20.0. Nextflow Sarek v2.7.1 workflow [20], included in nf-core v1.14 [21], was used to map the 150 bp and 100 X coverage paired-end sequences to the GRCh38/hg38 human reference genome. The alignment was carried out with BWA-MEM (Burrows–Wheeler Aligner MEM) [22] and the files were processed following the GATK (Genome Analysis Toolkit) [23] best practices using the following tools: MarkDuplicates, BaseRecalibrator, and ApplyBQSR. The SVs were called for each sample using the CNVkit Python library and the command-line software toolkit v0.9.8 [24]. The results were saved in segmented log2 ratios (CNS) format files and exported as Variant Call Format (VCF) files taking into account the sex of the individuals. Moreover, the Manta tool [25] was run using the Nextflow Sarek v2.7.1 workflow [20] included in nf-core v1.14 [21], and TIDDIT v2.3.1 [26] was employed following its pipeline. After the variant calling, variants with a length greater than 100,000 bp were ruled out using bcftools [27]. The filtered variants were annotated using AnnotSV v3.3.1 [28] with the split option.

### 2.3. Variant Prioritization

The variants were independently prioritized with each tool (CNVkit, Manta, and TIDDIT), following three different paths. The variants were filtered using the American College of Medical Genetics and Genomics (ACMG) annotation to select those that are pathogenic, likely pathogenic, or with uncertain significance regarding their pathogenicity. The SVs carried by the individuals in the severe tinnitus subgroup were studied separately from those in the non-severe tinnitus subgroup. The regions with an overlap of at least 60% in two or more samples of the same subgroup were identified using the SVDB toolkit v2.8.1 [29]. The regions covering genes in the FrequentLy mutAted GeneS (FLAGS) list and the olfactory receptor genes were filtered [30,31]. SVs carried by at least two individuals in the same subgroup and covering genes unique for severe tinnitus patients were retained. Finally, only variants identified by at least two of the three tools used were selected.

### 2.4. Public Database Annotation

To study the presence of SVs overlapping with the candidate variants of individuals with severe tinnitus in the reference populations, the Genome Aggregation Database (gnomAD) SVs v4.0.0 dataset was used for the global population (n = 807,162) and the Spanish Copy Number Alteration Collaborative Server (SPACNACS) database was used for the Spanish population (n = 417) [32]. In addition, constraints were evaluated for the genes covered by candidate SVs to measure their tolerance to variation. The loss-of-function observed/expected upper bound fraction (LOEUF) and the probability of being loss-of-function intolerant (pLI) were obtained from the gnomAD v4.0.0 dataset.

To evaluate the expression of the genes with candidate variants in the brain and inner ear, the following datasets were retrieved:RNA-Seq in human brain tissues: Amygdala, Anterior cingulate cortex (BA24), Caudate (basal ganglia), Cerebellar Hemisphere, Cerebellum, Cortex, Frontal Cortex (BA9), Hippocampus, Hypothalamus, Nucleus accumbens (basal ganglia), Putamen (basal ganglia), Spiral cord (cervical c-1), and Substantia nigra. The gene Transcripts per million (TPMs) were obtained from the Genotype-Tissue Expression (GTEx) project V8 [33].RNA-Seq in postnatal day 0 (P0) mouse hair cells and non-hair cells from the cochlea [34] from the gene Expression Analysis Resource (gEAR) portal (https://umgear.org, accessed on 17 November 2023). The reads per kilobase of transcript (RPKMs) from the orthologous genes were extracted.RNA expression by microarray in P0 mouse spiral ganglion neurons (SGNs) [35] from the Shared Harvard Inner-Ear Laboratory Database (SHIELD, https://shield.hms.harvard.edu, accessed on 17 November 2023). The expression levels were obtained from the orthologs of the candidate genes.

The expression values of each dataset were normalized between 0 and 100, with 100 being the maximum value of each database. The three databases were merged by gene name. The results were represented in a heatmap, scaling the values using the base-10 logarithm, using the ComplexHeatmap R package [36].

## 3. Results

### 3.1. Candidate Variants

All individuals were classified according to their THI score to identify a group of patients with a homogenous level of annoyance associated with tinnitus. The 75 individuals with a THI score above the third quartile (THI = 68) comprised the severe tinnitus subgroup, whereas the non-severe tinnitus subgroup included 235 individuals (Appendix A). Using the CNVkit, 244 variants were identified, 76 with Manta and 14,490 with TIDDIT for the severe subgroup before any filter was applied. After filtering, the following variants remained as candidates specifically associated with severe tinnitus using each tool: 30 variants in 66 genes carried by 31 individuals using CNVkit, 6 variants in 10 genes carried by 9 individuals using Manta, and 75 variants in 63 genes carried by 75 individuals using TIDDIT (Figure 1).

Only four candidate variants in eight genes were identified by two different tools, in both cases by Manta and TIDDIT.

### 3.2. Structural Variants Shared between Individuals with MD and Severe Tinnitus

Three heterozygous deletions were found in the same region of the *ERBB3* gene shared between the two unrelated individuals, I4-40 and I4-41 (Table 1). After examining the findings obtained by Manta and TIDDIT, three SVs covering the same region were found. However, there was a discrepancy in the variant type; Manta annotated the SVs as deletions and TIDDIT as duplications (Appendix A). Regarding this inconsistency, it was necessary to check the two samples in IGV (Figure 2), and three deletions were noted covering the reported positions by both tools. Furthermore, a heterozygous overlapping duplication was found in two unrelated individuals, I4-28 and I4-37 (Table 1), covering the genes *AP4M1*, *COPS6*, *MCM7*, *MIR106B*, *MIR25*, *MIR93*, and *TAF6*. In comparing the results obtained by both tools (Appendix A), they only varied at the start (one base of difference) and end positions (70 bases of difference). Despite this discrepancy between the tools, most of the SVs covered the same genome region. Moreover, both tools reported the duplication in the same individuals and the sequence was validated in IGV for both of the samples (Figure 2). The three deletions and the large duplications were checked in control individuals without the SVs (Appendix A). The pathogenicity of the three deletions and the duplication were uncertain according to the ACMG criteria (Appendix A).

Both individuals who shared the same three deletions in the *ERBB3* gene noted a THI of 80. I4-40 was a 64-year-old man who was diagnosed at the age of 50, and I4-41 was a 58-year-old woman diagnosed at the age of 52. Patient I4-40 suffered from unilateral SNHL and bilateral tinnitus, whereas I4-41 had bilateral SNHL and unilateral tinnitus.

Interestingly, both patients with the large duplication reported a THI score of 76. Patient I4-28 was a 72-year-old woman who was diagnosed with definite MD at the age of 22. Patient I4-37 was a 76-year-old woman diagnosed with definite MD when she was 30 years old. Moreover, both reported bilateral SNHL and tinnitus and suffered from migraines.

Although we cannot confirm the familial segregation, there was a familial history of MD in patient I4-40. Moreover, the other three patients also reported a familial history of SNHL.

### 3.3. Candidate Variants in the Reference Populations

The presence of these SVs was investigated in the reference populations. In the Spanish population from the SPACNACS database, no SVs covering the region were found. Nevertheless, in the gnomAD dataset, four deletions were found overlapping one or more regions of the candidate SVs in *ERBB3* for the individuals with severe tinnitus (Table 2). In addition, three duplications were identified, which overlapped part of the candidate duplication region (Table 2).

The SVs NC_000012.12:g.56100029_56100167del, NC_000012.12:g.56100244_56101058del, and NC_000012.12:g.56101363_56101526del found in the gnomAD database cover almost the same region as the three candidate deletions found in *ERBB3* from the individuals with severe tinnitus. The SV NC_000007.14:g.100076358_100093492dup overlaps part of the candidate duplication for severe tinnitus, NC_000007.14:g.100056545_100214180dup is much larger and covers the total length of the candidate variant.

Among the seven genes found within the duplication, only the genes *AP4M1*, *COPS6*, *MCM7*, and *TAF6* encode proteins. Regarding the constraint of those genes and *ERBB3*, the gene with the greatest constraint (the most intolerant to variants) was *COPS6* because it has a lower LOEUF score and a pLI score equal to 1 (Table 3).

### 3.4. Expression of Genes with Candidate Variants in the Brain and Inner Ear

The *ERBB3* gene was highly expressed in the cochlea, brain, and SGN. Three of the four coding genes covered by the large duplication were expressed in all the studied tissues, especially the *COPS6* gene, whereas *AP4M1* showed less expression in the brain and cochlear tissues and was not expressed in the SGN. However, none of the three non-coding genes were expressed in the studied tissues (Figure 3).

## 4. Discussion

Severe tinnitus is a rare, debilitating condition associated with otological and non-otological risk factors, such as SNHL, depression, and sleep disorders [8,37]. In most cases, tinnitus also co-exists with hyperacusis, a reduced tolerance to loud noise observed in between 8% and 15.2% of the population [38]. Interestingly, 80% of individuals with severe tinnitus also suffer from hyperacusis [39], but the molecular links between severe tinnitus and hyperacusis have not been established. While vertigo attacks are noted as the primary symptom during the initial years in individuals suffering from MD, severe tinnitus is reported as the most troublesome symptom by many patients when episodes of vertigo are scarce over time [40,41].

The genetic structure of tinnitus is complex, with common and rare variants contributing to different effect sizes in the phenotype. Severe tinnitus is associated with rare missense variants in 24 synaptic genes [9]. In the present work, the exome sequencing data of the MD cohort has been extended by investigating SVs. We report the first CNVs observed in patients with MD and severe tinnitus in several unrelated patients: three deletions in the *ERBB3* gene and a large duplication involving multiple genes.

Although several rare missense SNVs were reported in MD and severe tinnitus, including *ANK2*, *TSC2*, and *AKAP9* genes [9], these variants do not explain the presence of other comorbidities with a higher prevalence. Therefore, discovering large SVs that include coding and non-coding regions with multiple rare and common variants can enhance the molecular understanding of a complex multidimensional phenotype associated with emotional, cognitive or behavioral dysfunction [1,8]. This type of variation could also explain the different facets observed in severe tinnitus, with substantial interindividual heterogeneity in the phenotype in terms of associated comorbidities, such as hyperacusis, anxiety, depression, or headaches.

The large duplication was identified in two individuals with early-onset MD, with an age of onset of 22 years and 30 years for I4-28 and I4-37, respectively. The difference in the age of onset is remarkable when comparing individuals I4-40 and I4-41, which carried the three deletions in *ERBB3.* The importance of finding biomarkers for understanding the prognosis, outcomes, and therapeutic response of other complex diseases in early-onset individuals, such as multiple sclerosis, has been demonstrated [42]. Furthermore, both patients carrying the large duplication reported migraines. It was previously observed that MD patients with migraines have an earlier age of onset of disease compared to those without migraines [43,44].

Three rare deletions were found in two individuals in the *ERBB3* (erb-b2 receptor tyrosine kinase 3) gene, which encodes for the ERBB3 (receptor tyrosine-protein kinase erbB-3) protein. In rats, the Erbb3 receptor is located in the nucleus and cytoplasm of Schwann cells, which are in charge of the transduction of Neuregulin-1 (NRG-1) growth factor [45]. Furthermore, in mice, the Erbb3 protein is fundamental to the myelination and migration of the Schwann cells [46]. Therefore, it is essential for *Schwann cell differentiation* (GO:0014037), according to the Gene Ontology (GO) database [47,48]. It is the only gene participating in the *absent Schwann cell precursors* (MP:0001109), and is related to *abnormal myelination* (MP:0000920) based on the Mouse Genome Informatics (MGI) database [49]. The Schwann cells are the myelinating cells of the peripheral nervous system. Myelination increases the conduction velocity along the axon. Moreover, these cells are necessary in the axon damage response and their regeneration in the peripheral nervous system [50], a phenomenon occurring in noise-induced hearing loss and age-related hearing loss [51]. In addition, minor injury to the auditory nerve leads to different outcomes for neurons in the cochlear nucleus, such as increased excitation, which is suggested to be associated with tinnitus [52,53].

A large duplication covering seven genes was also found in two individuals. The extremes of the SV were located in the *TAF6* (TATA-box binding protein associated factor 6) and *COPS6* (COP9 signalosome subunit 6) genes. The protein TAF6, encoded by the *TAF6* gene, is one of the principal components of the TATA-box-binding protein (TBP), which is essential for initiating gene transcription by RNA polymerase II [54]. It is associated with the Cornelia de Lange and Alazami-Yuan syndromes [55,56]. Interestingly, the *TAF6* gene was also enriched in missense variants in a Swedish cohort, composed of 147 patients with chronic and constant tinnitus [10]. In addition, the coding genes *AP4M1* and *MCM7* are covered by the large duplication. Variants in *AP4M1* are related to spastic paraplegia, intellectual disability, hearing loss, and microcephaly [57,58]. In individuals with diseases previously related to the *TAF6* and *AP4M1* genes, some of the clinical features observed include reduced growth, including microcephaly. This might suggest that malformations in the maxillo-temporal bone could trigger the occurrence of tinnitus. Interestingly, these individuals were diagnosed with early-onset MD. The protein MCM7, encoded by the *MCM7* (minichromosome maintenance complex component 7) gene, is part of the MCM hexameric complex. These proteins are essential for replication and cell cycle progression [59,60]. Three of the seven genes are non-coding; these are *MIR25*, *MIR93*, and *MIR106B*. All three genes were transcribed into microRNAs, sharing common biological processes, such as *gene silencing by miRNAs* (GO:0035195), and molecular functions, such as *mRNA binding involved in posttranscriptional gene silencing* (GO:1903231), as described by the GO database [47,48]. None of the microRNAs were found to target the candidate genes [61].

To study the inheritance of tinnitus, different approaches have been followed. The identification of SNVs through genome-wide association studies (GWASs) is noteworthy. Four different GWASs were performed for this trait [62,63,64,65]. These studies identified two common variants in the Chinese population (rs2846071 and rs4149577) in the intron of *TNFRSF1A*, which were found to be associated with noise-induced tinnitus [63]. However, most of the GWASs have not been replicated, demonstrating the limitations of this method with rare disorders and the poorly-defined phenotypes associated with tinnitus, hearing loss, and other common associated comorbidities in the Biobank database [66].

In this work, the exomes of the patients were sequenced, which covers only the coding regions. It would be necessary to perform whole-genome sequencing to study the non-coding regions, which play an essential role in gene expression. Moreover, some genes that overlap with the large duplication are associated with multiple functions, necessitating further analysis to confirm which gene or genes are related to severe tinnitus.

## 5. Conclusions

The following conclusions were derived from this work:Rare SVs were found in several unrelated individuals with MD and severe tinnitus;Three deletions in the *ERBB3* gene were identified in two individuals with severe tinnitus;A large duplication overlapping the candidate genes *AP4M1*, *COPS6*, *MCM7*, and *TAF6* and the non-coding genes *MIR106B*, *MIR25*, and *MIR93* was carried by two MD patients with severe tinnitus;These findings support the role of large SVs in shaping the genetic architecture of severe tinnitus in MD and define new candidate genes associated with this phenotype.

## Figures and Tables

**Figure 1 genes-15-00022-f001:**
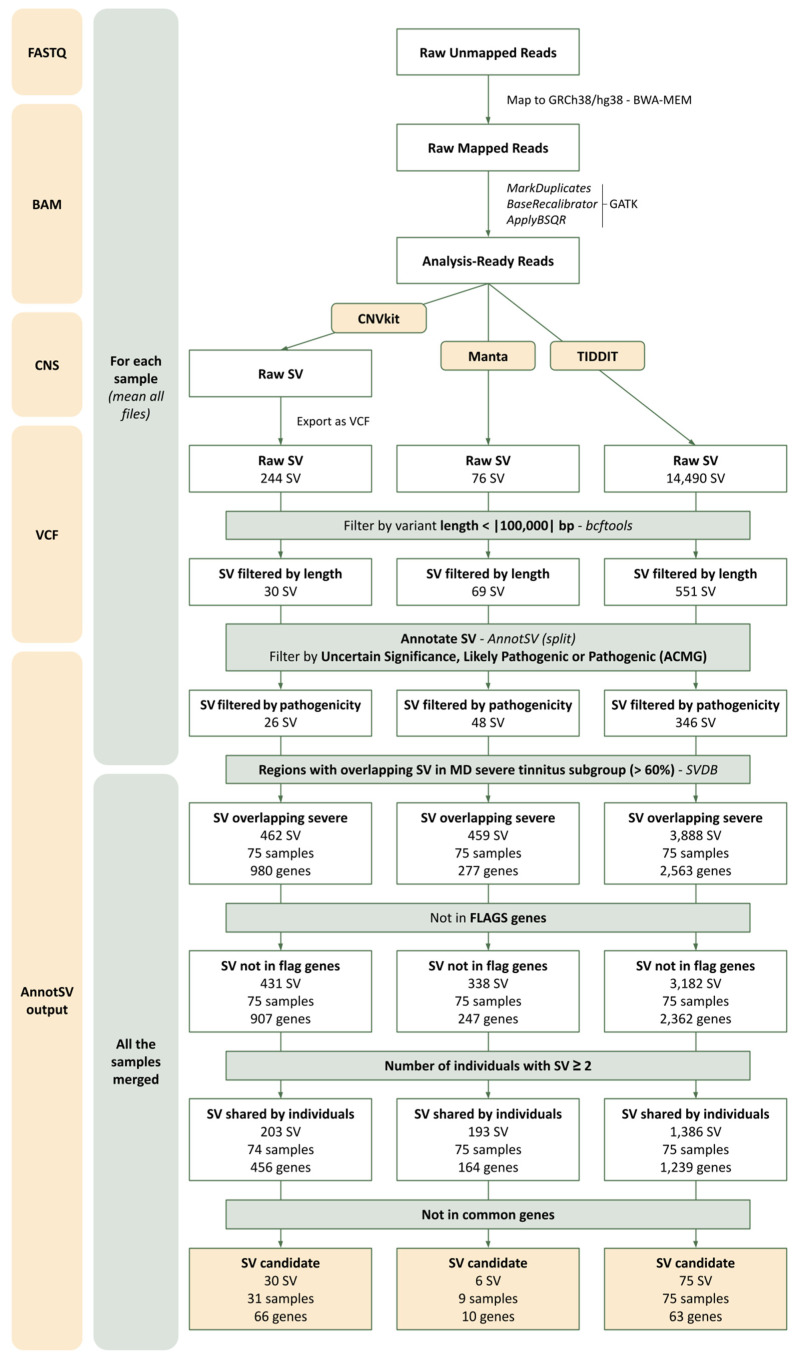
Flowchart summarizing the identification and prioritization strategy for structural variants (SVs) from exome sequencing datasets in Meniere Disease (MD) patients with severe tinnitus. The variants were obtained using the CNVkit, Manta, and TIDDIT tools. Each box shows the number of remaining SVs after each filter. In addition, after SV annotation, the number of samples carrying these SVs and the number of genes with overlapping SVs were indicated. BAM: Binary Alignment Map; CNS: Segmented log2 ratios; VCF: Variant Call Format; BWA-MEM: Burrows–Wheeler Aligner MEM; GATK: Genome Analysis Toolkit; bp: base pair; ACMG: American College of Medical Genetics and Genomics; FLAGS: FrequentLy mutAted GeneS.

**Figure 2 genes-15-00022-f002:**
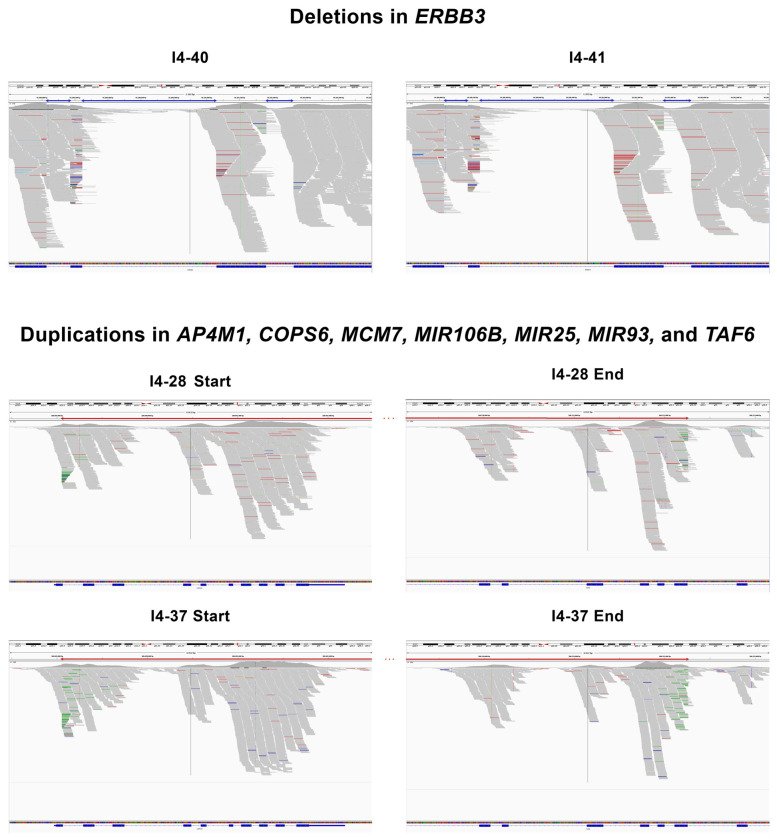
Validation of the deletions in the *ERBB3* gene in individuals I4-40 and I4-41 in IGV and the large duplication in *AP4M1*, *COPS6*, *MCM7*, *MIR106B*, *MIR25*, *MIR93*, and *TAF6* genes in individuals I4-28 and I4-37. The deletions are indicated by blue arrows and the duplications are indicated by red arrows.

**Figure 3 genes-15-00022-f003:**
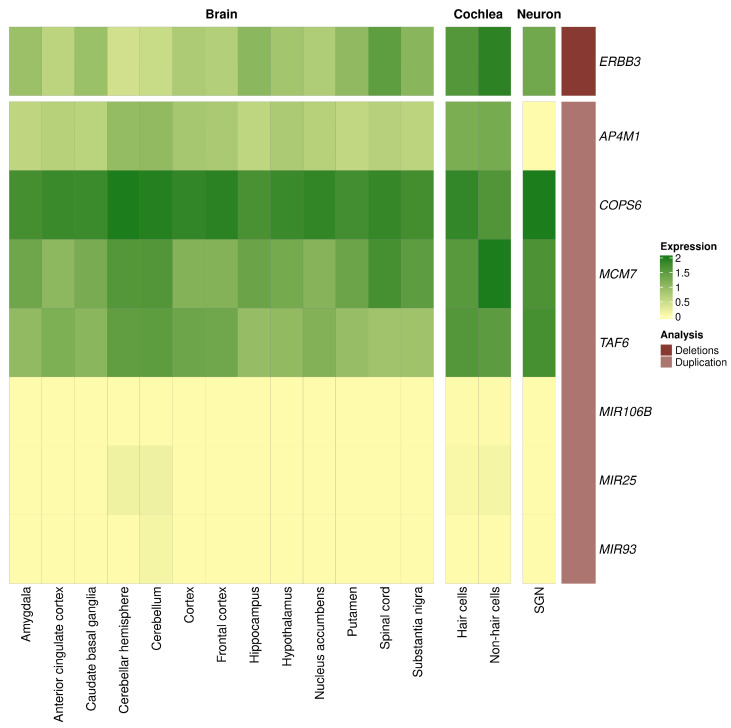
Expression of the genes covered by candidate variants for individuals with severe tinnitus. The heatmap shows the expression levels of the genes in 13 brain tissues obtained from GTEx in humans, the expression levels of the orthologous genes in the cochlear hair cells and non-hair cells from gEAR in postnatal day 0 (P0) mice, and the expression levels of the genes in the spiral ganglion neuron (SGN) from SHIELD in P0 mice. The expression of each dataset was normalized from 0 to 100 and is represented as the base-10 logarithm. Each gene was labelled by the type of structural variant overlapping the gene.

**Table 1 genes-15-00022-t001:** Summary of structural variants (SVs) found in individuals with severe tinnitus using Manta.

Chr	Start	End	Length (bp)	SV Type	Individuals	Gene Symbol	ACMG
12	56,100,028	56,100,172	144	Deletion	I4-40, I4-41	*ERBB3*	US
12	56,100,243	56,101,058	815	Deletion	I4-40, I4-41	*ERBB3*	US
12	56,101,359	56,101,526	167	Deletion	I4-40, I4-41	*ERBB3*	US
7	100,089,053	100,112,257	23,204	Duplication	I4-28, I4-37	*AP4M1*, *COPS6*, *MCM7*, *MIR106B*, *MIR25*, *MIR93*, *TAF6*	US

Chr: chromosome; ACMG: American College of Medical Genetics and Genomics; US: uncertain significance.

**Table 2 genes-15-00022-t002:** Summary of structural variants listed in the gnomAD dataset overlapping the genomic coordinates of the candidate regions for individuals with severe tinnitus.

Chr	Start	End	Length (bp)	SV Type	gnomAD NFE		gnomAD	
					Frequency	Number of Individuals	Frequency	Number of Individuals
12	56,100,029	56,100,167	138	Deletion	1.02 × 10^4^	6	8.74 × 10^−5^	11
12	56,099,811	56,101,058	1247	Deletion	1.69 × 10^5^	1	2.38 × 10^−5^	3
12	56,100,244	56,101,058	814	Deletion	1.36 × 10^4^	8	1.27 × 10^−4^	15
12	56,101,363	56,101,526	163	Deletion	1.86 × 10^4^	11	1.35 × 10^−4^	14
7	100,076,358	100,093,492	17,134	Duplication	3.39 × 10^5^	2	3.17 × 10^−5^	4
7	100,056,545	100,214,180	157,635	Duplication	0	0	7.93 × 10^−6^	1
7	100,095,604	100,095,982	378	Duplication	0	0	7.93 × 10^−6^	1

Chr: chromosome; SVs: structural variants; gnomAD NFE: Non-Finnish European for gnomAD; gnomAD: global population for gnomAD.

**Table 3 genes-15-00022-t003:** Estimation of constraints for *ERBB3*, *AP4M1*, *COPS6*, *MCM7*, and *TAF6* genes.

Gene Symbol	LOEUF	pLI
*ERBB3*	0.685	0
*AP4M1*	1.107	0
*COPS6*	0.374	1
*MCM7*	1.307	0
*TAF6*	0.657	0.55

LOEUF: loss-of-function observed/expected upper bound fraction; pLI: probability of being loss-of-function intolerant.

## Data Availability

The data presented in this study are available on request from the corresponding author. The data are not publicly available due to the dataset containing unique variants associated with particular individuals with Meniere Disease. This restriction is in accordance with the European Regulations on private data protection to prevent participant identification, following Regulation (EU) 2018/1725.

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
