# Peer review of "Rare Deletions or Large Duplications Contribute to Genetic Variation in Patients with Severe Tinnitus and Meniere Disease"

_genes, 2023, doi:10.3390/genes15010022_

Round 1
Reviewer 1 Report
Comments and Suggestions for Authors
This manuscript “Rare deletions and a large duplication in patients with severe tinnitus and Meniere Disease” by Escalera-Balsera et al. discussed a few rare structural variants at two genomic loci, one with three deletions and the other one with a large duplication. The authors explained in details how they selected the patients, performed whoe exon sequencing, analyzed the data with multiple packages, filtered the raw hits with several rounds of filtration, and identified those structural variants. The authors also checked the expression of potential affected genes against gEAR portal and found that those genes, particularly protein coding genes, are expressed in neurons and cochlear cells. Based on those pieces of data, they concluded that they identified a few rare structural variants in patients and those variants potentially affect the genes and the normal function of the inner ear. Whole exon sequencing has been demonstrated as a powerful tool to identify pathogenic variants, and its application in the tinnitus and Meniere Disease patients definitely will help to find genes involved in those diseases and would be of great interest to both scientists and clinicians. However, I am not quite convinced by the data showing here.
First of all, three deletions at the ERBB3 locus with three introns perfectly, which requires a justification with control data to show that it is not due to cDNA contamination. Second, the IGV signal tracks for the large duplication do not demonstrate the “duplication” by significantly higher coverage, at least at the loci of start and end regions. Why were those deletions and duplication not called by CNVkit package? Because those variants were identified based on split-reads but not by coverage difference? If it’s based on split reads, then the IGV track should show such reads, otherwise should show the coverage difference. I am not confident to take the recalls by 2 out of 3 packages as real variants by the data shown here. Should try more packages, or provide some experimental data to support the conclusion.
In addition to the major concern, there are a few other minor issues:
11. The workflow chart was cut into two parts, one in the main figure and the other part in supplemental figures, which will confuse readers.
22. Please state the version of reference genome assembly since genomic coordinates were used
33. Regions of deletion and duplication should be clearly labeled on IGV tracks. Actually, should move IGV tracks from supplemental data to main figures.
44. Spell out SNV at least once.
Comments on the Quality of English LanguageNA
Reviewer 2 Report
Comments and Suggestions for Authors
Regarding the title, the suggestion is to review it, as in this research only four patients out of an extensive cohort were identified with molecular alterations, either a deletion or a rare duplication, and not with both alterations as the present title implies. Also, the title suggests the alterations are a main cause for the disease in general, in this subject and regarding the discussion, they may be considered by the authors as a contributing factor to the presentation of the disease as they were not identified in the other patients with a similar clinical presentation.
Regarding the patients, it would be of interest to discuss, if possible, if the patients were the only individuals in their given family with the disease.
It would be interesting to discuss the difference between the identified variants regarding the clinical presentation, as described, there were decades of difference in the clinical presentation/diagnosis of the patients depending on the molecular alteration and to discuss how this situation relates to the implied gene or genes affected.
In the discussion, the affected genes are described regarding its known function, etc., but it would be interesting to clarify by the authors how they considere that this given function is related/contributes to the presentation of the disease. This information and the final part of the abstract should be considered in the conclusions.
It may add to the information provided, that Meniere disease is classified in the OMIM data base with the number 156000, and also to confirm by literature review that all previously identified genes related to this disease, particularly in the Spanish population, were not identified in this analysis.
Comments on the Quality of English LanguageReview the title as it implies that both alterations were present simultaneously in all the patients and it may infer that they were a frequent cause.
Reviewer 3 Report
Comments and Suggestions for Authors
Escalera-Balsera et al reported deletions and duplication responsible for severe tinnitus and Meniere disease. This is an interesting article. However, I did have some difficulty to understand the Figure 1. The text is inadequate, and there are very few explanations. For example, it's difficult to understand how 30 "variants" can be present in 66 genes. I think the term "variant" is inappropriate. It should be specified that these are large deletions or duplications or just replace variants with structural variant (SV) as mentioned in figure 1. The authors should provide the list of deletions and duplications with the genes involved.
Line 158-159: write: only four candidate SV in eight genes.....
There is no indication about patient status: heterozygous, homozygous? Is it carried by the same allele?
There is an error in Table 2: Individ-Indi. I think it's Individuals.
The deletions and duplication identified in the study have already been reported. Is there any information on these patients or any indication of a possible phenotype?
Is the statistical significance sufficient? Can we be certain that these SVs are responsible for severe tinnitus when the other individuals identified with these SVs have not been reported with severe tinnitus?
Round 2
Reviewer 1 Report
Comments and Suggestions for Authors
All my concerns were addressed.